# Crosslinked Bifunctional Particles for the Removal of Bilirubin in Hyperbilirubinemia Cases

**DOI:** 10.3390/ma16082999

**Published:** 2023-04-10

**Authors:** María del Prado Garrido, Ana Maria Borreguero, Maria Jesús Ramos, Manuel Carmona, Francisco Javier Redondo Calvo, Juan Francisco Rodriguez

**Affiliations:** 1Department of Chemical Engineering, Institute of Chemical and Environmental Technology, University of Castilla-La Mancha, Avda. De Camilo José Cela 1, 13005 Ciudad Real, Spain; mariaprado.garrido@uclm.es (M.d.P.G.); anamaria.borreguero@uclm.es (A.M.B.); mariajesus.ramos@uclm.es (M.J.R.); manuel.cfranco@uclm.es (M.C.); 2Department of Anesthesiology and Critical Care Medicine, University General Hospital, Obispo Rafael Torija s/n, 13005 Ciudad Real, Spain; fjredondo@sescam.jccm.es; 3Faculty of Medicine, University of Castilla-La Mancha, Camino de Moledores s/n, 13005 Ciudad Real, Spain

**Keywords:** crosslinking, suspension polymerization, BSA immobilization, bilirubin removal, wetting effect

## Abstract

This work describes the development of styrene-divinylbenzene (St-DVB) particles with polyethylene glycol methacrylate (PEGMA) and/or glycidyl methacrylate (GMA) brushes for the removal of bilirubin from blood in haemodialyzed patients. Bovine serum albumin (BSA) was immobilized onto the particles using ethyl lactate as a biocompatible solvent, which allowed the immobilization of up to 2 mg BSA/g of particles. The presence of albumin on the particles increased their capacity for bilirubin removal from phosphate-buffered saline (PBS) by 43% compared to particles without albumin. The particles were tested in plasma, finding that St-DVB-GMA-PEGMA particles that had been wetted in ethyl lactate with BSA reduced the concentration of bilirubin in plasma by 53% in less than 30 min. This effect was not observed in particles without BSA. Therefore, the presence of albumin on the particles enabled quick and selective removal of bilirubin from plasma. Overall, the study highlights the potential use of St-DVB particles with PEGMA and/or GMA brushes for bilirubin removal in haemodialyzed patients. The immobilization of albumin onto the particles using ethyl lactate increased their capacity for bilirubin removal and enabled quick and selective removal from plasma.

## 1. Introduction

Hyperbilirubinemia is indeed a serious medical condition that can have life-threatening consequences for patients with liver dysfunction. The accumulation of excess bilirubin in the blood can lead to a range of symptoms, including jaundice, fatigue, and abdominal pain. In severe cases, the buildup of bilirubin can lead to brain damage and death [1,2]. Bilirubin exists in three different states in the human body. Direct or conjugated bilirubin is formed in the liver when bilirubin is conjugated, or attached, to glucuronic acid. This form of bilirubin is water soluble and can be excreted in the bile and ultimately eliminated from the body. Non-conjugated bilirubin, also known as unconjugated bilirubin, is the form of bilirubin that is not attached to glucuronic acid. Instead, it binds to albumin in their primary binding sites with high affinity (with a constant of 1.4·10^7^ L/mol [3]) and even in secondary sites with lower affinity [4], being finally transported to the liver for processing. Free bilirubin is not bound to either glucuronic acid or albumin and is therefore not water soluble. As a result, it can accumulate in tissues, causing damage and leading to hyperbilirubinemia. Free bilirubin is the most dangerous due to its insolubility in plasma [5]. It is therefore crucial to develop new materials for the removal of bilirubin from the blood, especially for patients with liver dysfunction.

Liver transplantation is currently the most effective treatment for hyperbilirubinemia in cases where other treatments have failed. However, the availability of donors is limited, and the procedure itself can be risky and expensive. Hemodialysis [6] is a treatment option that can help remove bilirubin from the blood, but it is not selective and can also remove other important substances from the blood. Another treatment option also based in hemodialysis is the Molecular Adsorbent Recirculating System (MARS^®^), which is a form of extracorporeal liver support that can selectively remove bilirubin and other toxins from the blood. However, MARS^®^ treatment is expensive, complex [7], and can cause coagulopathies in some patients [8].

Most of the alternatives being explored for the treatment of hyperbilirubinemia involve the use of particulate materials for the hemodialysis process. Some of the commonly used materials include poly(styrene-methyl methacrylate) [7,8], polystyrene-divinylbenzene [9,10] polyvinyl alcohol gels [11], or polyethersulphone particles [12]. However, their current selectivity for bilirubin removal is limited. Previous work has also explored the use of particles based on styrene (St) and methyl methacrylate (MMA) for bilirubin removal [13]. However, these particles experience a large swelling effect when they come into contact with blood, which can lead to blood coagulation and make them unsuitable for use in patient treatment.

One solution to the swelling problem is to incorporate crosslinkers during the synthesis of the particles. This creates a three-dimensional network that promotes stiffer polymers and improves their mechanical stability. The best way to avoid or at least reduce the problem of swelling is the use of chemical compounds able to link the polymer chains to form a stable tridimensional network that avoids the swelling or shrinking of the materials. These crosslinkers have been widely described previously in the literature for the synthesis of polymeric particles [14,15,16]. Some of the crosslinkers previously described in the literature are ethylene glycol dimethacrylate (EGDMA) [17,18,19], *N*,*N*-methylenebis-acrylamide (NNMBA) [20], or cyclodextrin [21]. However, the most common crosslinker for styrene-based particles is divinylbenzene (DVB) [22,23,24,25]. For that reason, DVB was selected in this work.

Thus, this research proposes the synthesis of new crosslinked particles of poly(styrene-co-divinylbenzene) functionalized with PEGMA and/or GMA brushes by means of suspension polymerization. This functionalization allows us to more easily link of the protein (albumin) to the polymer [26], while the crosslinker (DVB) reduces the swelling effect that can cause the formation of a gel structure when particles get wet, minimizing the coagulation of blood in the particle bed that could occur when passing through the gel matrix. The synthesized particles were then pre-wetted with ethyl lactate, which improved albumin immobilization on the particles [14]. The properties of the developed particles are suitable for application in the traditional cartridges of hemodialysis and be used for hyperbilirubinemia treatment.

## 2. Materials and Methods

### 2.1. Materials

Sigma-Aldrich (St. Louis, MO, USA) provided all the reagents necessary for the synthesis of the particles, including St, DVB, PEGMA, and GMA as monomers, benzoyl peroxide (BPO) as an initiator, and polyvinylpyrrolidone (PVP) as dispersant agent. Ethyl lactate (EL) used for the swelling of the particles was also provided by Sigma-Aldrich. Additionally, Panreac (Barcelona, Spain) supplied the salts required for the synthesis of phosphate-buffered saline (PBS, 0.01 M phosphate buffer, pH 7.4), while the human plasma was obtained from the blood bank of the University General Hospital of Ciudad Real (Ciudad Real, Spain). Finally, Linear (Barcelona, Spain) supplied the Cromatest kit for determining BR in plasma.

### 2.2. Methods

#### 2.2.1. Synthesis of Polymeric Particles

The suspension polymerization reactions took place in a 2 L jacketed glass reactor. The synthesis was divided into two phases: a continuous phase consisting of water and the suspending agent, and a discontinuous phase consisting of monomers and the initiator. The reaction formulations used for the experiments are presented in Table 1.

Initially, the reactor was charged with water and PVP, and agitated at 400 rpm while heating to 60 °C. Subsequently, the discontinuous phase was introduced into the continuous phase. For the synthesis of St-DVB particles, the polymerization process was carried out for 6 h at 70 °C. In contrast, the reactions for functionalized particles (St-DVB-PEGMA, St-DVB-GMA, and St-DVB-GMA-PEGMA) were performed at 80 °C for 6 h. After completion, the particles were washed with ethanol and air dried for 72 h at room temperature.

#### 2.2.2. Albumin Immobilization

The immobilization of albumin was performed using an Erweka USP Flow Through Cell Apparatus (Hessen, Germany). Firstly, a specific weight of particles (Wp) was pre-wetted in EL and placed into the cells. Next, a BSA solution with a concentration of 0.2 mg/mL in PBS was passed through the particles at a flow rate (Fr) of 8 mL/min for a duration time (t) of 72 h at a temperature (T) of 25 °C.

#### 2.2.3. Bilirubin Removal

To study the bilirubin removal kinetic and capacity, the desired amount of bilirubin was dissolved in 0.06 M sodium hydroxide and added to either PBS or plasma. The resulting mixture was then passed through a bed of particles at a flow rate of 120 mL/min for a period of 2 h. The conditions were selected according to conventional protocols for extracorporeal depuration treatments [15]. The experiments were conducted at room temperature and protected from light to prevent degradation of bilirubin.

### 2.3. Characterization of Polymers

#### 2.3.1. Scanning Electron Microscopy

The morphology of the synthesized particles was observed by using a Quanta 250 scanning electron microscope (FEI Company, Hillsboro, OR, USA).

#### 2.3.2. Particle Size and Polydispersity Index

The measurements of particle size and the distribution width (Span) were carried out dispersing samples of the polymer microparticles in air by low-angle laser light scattering (LALLS) using a Malvern Mastersizer 2000 (Cambridge, UK) equipped with a Scirocco 2000 unit.

#### 2.3.3. Fourier Transform Infrared Spectroscopy (FT-IR)

Copolymer composition was studied by FT-IR. Infrared spectra were obtained by a Spectrum Two spectrometer (Perkin Elmer, Inc, Waltham, MA, USA) at room temperature.

#### 2.3.4. Epoxide Content

The particles’ epoxide content was determined by the standard titration method UNE-EN ISO 3001 [16].

#### 2.3.5. Protein Concentration Determination

The Lowry method [17] was used to quantify the amount of albumin immobilized in the polymer. The absorbance was measured at 750 nm using a UV-VIS apparatus (JASCO V-750) (Easton, MD, USA) provided with the software Spectra Manager.

#### 2.3.6. Bilirubin Concentration Determination

Bilirubin in PBS was measured directly at 450 nm, while bilirubin in plasma was measure with a kit Cromatest measuring at 540 nm. All the measurements were carried out in a UV-VIS apparatus (JASCO V-750).

## 3. Results

### 3.1. St-DVB Functionalized Particle Synthesis

Various types of particles were synthesized using St and DVB as base materials and functionalized with PEGMA and/or GMA. The reaction yields for these particles ranged between 60.8 and 96.5, as shown in Table 2. The table also includes their median diameter in volume (Dpv 0.5), the Span, and the estimated pressure drops (ΔP) calculated using the Ergun equation [18].

The experimental results indicate that the addition of PEGMA and/or GMA to St-DVB results in a reduction in particle size, with Dpv 0.5 values that are 11.5%, 53%, and 42% smaller for St-DVB-PEGMA, St-DVB-GMA, and St-DVB-GMA-PEGMA, respectively. These findings suggest that functionalization with GMA has a significant effect on particle size reduction. Previous studies have also reported that the use of GMA and PEGMA can reduce particle size [19,20]. However, in the case of St-DVB particles, the effect of GMA on particle size reduction is more pronounced. Despite the differences in particle size, all the synthesized particles are in the size range potentially suitable for use in hemodialysis cartridges [21]. In addition, the estimated pressure drops using the Ergun equation are within the limits for extracorporeal systems to prevent blood coagulation, for which maximum admissible pressure drop is 200 mmHg [22].

FT-IR spectroscopy was used to chemically characterize both functionalized and non-functionalized particles, as shown in Figure 1. A common adsorption band was observed at 3020 cm^−1^, which was attributed to the aromatic C-H stretching vibration. Moreover, peaks in the range of 1600–1440 cm^−1^ were observed, which corresponded to the C=C phenyl stretching. The bands at 750 and 695 cm^−1^ were also observed, which corresponded to the aromatic C-H bending of benzene derivative from polystyrene and polydivinylbenzene, as previously reported [23,24].

The functionalization of the particles with epoxide group could not be demonstrated by FT-IR spectroscopy due to the overlay of the epoxide group with the characteristic peaks of the polymer. Thus, the epoxide content of the particles functionalized with GMA was determined, with values of 19 μmol/g particle for GMA-only functionalized particles and 10 μmol/g particle in the case of GMA and PEGMA functionalization.

SEM analysis was conducted to examine the morphology of the synthesized particles, as shown in Figure 2. In the case of St-DVB-PEGMA, the use of SDS resulted in smaller particles that formed irregularly shaped agglomerates. Although these particles were smaller than St-DVB-GMA and St-DVB-GMA-PEGMA, individual particle size could not be measured by LALLS analysis due to the agglomerate formation. On the other hand, particles synthesized with PVP had a predominantly spherical morphology.

### 3.2. BSA Immobilization

In order to link albumin to PEGMA [25] and/or GMA brushes, it is necessary to control the pH to prevent protein denaturation due to their sensitivity to pH changes. Therefore, physiological pH conditions (pH = 7.4) were employed, as proteins are highly sensitive towards pH changes [26].

The quantity of BSA immobilized onto the particles was determined through UV spectroscopy, by calculating the difference between the initial and final BSA concentration in PBS solution. The kinetics of albumin immobilization onto non-wetted St-DVB based particles was carried out (Figure 3). The BSA immobilized in each case was 1.8 mg BSA/g particles for non-functionalized particles, 1.3 mg BSA/g particles for functionalized particles with GMA and PEGMA, and 0.53 mg BSA/g particles for PEGMA functionalized particles. In the case of GMA functionalization, no BSA was immobilized onto the particles.

As previously reported by the research group [27], the wetting of the particles based on St-MMA in ethyl lactate implied a clear improvement in the albumin immobilization. For that reason, the albumin retention of these St-DVB particles will be also studied after prewetting in EL. The particles were pre-wetted with ethyl lactate before the immobilization process, as it facilitates access to the linking sites. The time evolution of BSA concentration is illustrated in Figure 4.

As is shown in Figure 4, the particles without functionalization represented the best support for albumin immobilization, with an average value of 2 mg BSA/g particles. PEGMA functionalization showed a similar performance, with 1.9 mg BSA/g particles while GMA and GMA-PEGMA functionalization presented values of only 1.5 and 1.25 mg BSA/g particles, respectively. The reduction of immobilized BSA in the functionalization with GMA can be explained due to its capacity as a slight crosslinker [27,28] close to the well-known use of divinylbenzene as a crosslinker [29]. The ability of GMA to act as a mild crosslinker can prevent the swelling effect, but it may also restrict access to certain binding sites. This leads to the formation of a highly crosslinked, rigid 3D structure that can impede the entry of large biomolecules like albumin into the internal active sites, ultimately decreasing the capacity for albumin immobilization on the particles.

### 3.3. Bilirubin Removal Tests

In order to assess the potential application of these materials as an alternative treatment for hyperbilirubinemia, in vitro tests were conducted to evaluate their capacity to remove bilirubin in two different media (PBS and human plasma).

The kinetics of bilirubin removal onto non-wetted St-DVB particles with or without BSA were determined (Figure 5). The best bilirubin uptake results were found for the particles functionalized with GMA and PEGMA, with 0.27 and 0.52 mg BR/g particles for the cases without and with BSA, respectively. In the case of particles without BSA, particles without functionalization and functionalized with PEGMA removed about 0.15 mg BR/g particles. In the case of particles functionalized with GMA, the result was only 0.05 mg BR/g particles, improving a little by the incorporation of BSA, up to 0.22 mg BR/g particles. In the case of particles functionalized with PEGMA or non-functionalized and containing BSA, no bilirubin was retained in the polymers.

In order to improve the results, the particles were pre-wetted in ethyl lactate and tested with and without BSA for bilirubin removal from PBS media. The progress of bilirubin removal over time was recorded and is presented in Figure 6, while the final bilirubin uptake values are summarized in Table 3.

According to these results, the particles with BSA showed a faster kinetic for removing bilirubin since, while particles without BSA needed 90 min or more to reach the equilibrium, the particles with BSA needed less than one hour to reach the end point. Besides, the final BR uptake was up to 43% higher when BSA was attached (Table 3). In all cases, the incorporation of PEGMA and GMA improved the capacity for BR removal, but this effect is higher in the case of particles without BSA. Thus, the BR removal is affected in the presence of BSA but also by the PEGMA and GMA functionalization, probably because they form branches that facilitate the link with the BR.

Finally, the ability of the particles to remove bilirubin from plasma was investigated. The initial concentration of bilirubin was set at 7.5 mg/dL, which is commonly found in patients with hepatic failure. It should be noted that, unlike in PBS, human plasma contains not only free bilirubin, but also conjugated or direct bilirubin (dBR) and non-conjugated or indirect bilirubin (iBR), resulting in a higher total initial bilirubin concentration in human blood. St-DVB-GMA-PEGMA particles previously wetted in ethyl lactate were chosen due to their superior BR uptake from PBS. To confirm the importance of albumin immobilization in BR removal, experiments were also conducted with wetted St-MMA-GMA-PEGMA without BSA under the same conditions.

According to the results in Figure 7a, particles that did not contain BSA were not effective in removing bilirubin from plasma. On the contrary, Figure 7b shows the results of two experiments with the same initial bilirubin concentration, with the particles removing an average of 8.5 mg of bilirubin per gram of particles in less than 30 min. The amount of indirect bilirubin removed in each experiment was 8.3 and 8.75 mg BR/g of particles, respectively. Thus, the presence of albumin provided selectivity, preventing the particles from St-MMA-GMA-PEGMA becoming saturated with other blood components or metabolites.

## 4. Conclusions

St-DVB particles functionalized with PEGMA and/or GMA brushes were created with sizes ranging from 460 to 868 μm, making them suitable for use in standard hemodialysis cartridges. The St-DVB-GMA-PEGMA particles had a theoretical pressure drop of up to 9 mmHg, which is 95.5% lower than the maximum pressure drop acceptable for extracorporeal systems (200 mmHg). BSA immobilization levels varied among the particles, with values of 2, 1.9, 1.5, and 1.25 mg/g achieved for St-DVB, St-DVB-PEGMA, St-DVB-GMA, and St-DVB-GMA-PEGMA, respectively. Despite this, St-DVB-GMA-PEGMA particles with BSA demonstrated the highest bilirubin removal capacity, achieving a 53% reduction in plasma bilirubin levels. In contrast, particles without BSA were not able to remove bilirubin from plasma, indicating that albumin immobilization was key to selectively reducing bilirubin levels.

## Figures and Tables

**Figure 1 materials-16-02999-f001:**
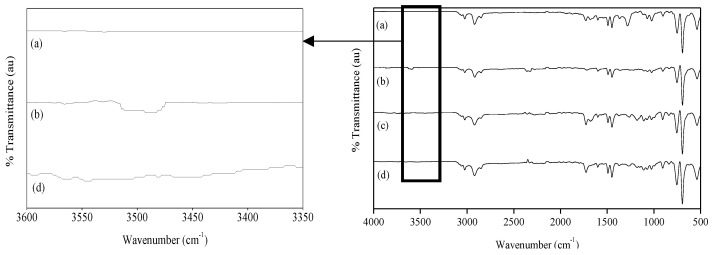
FT-IR spectra of the different synthesized particles: St-DVB (a), St-DVB-PEGMA (b), St-DVB-GMA (c), and St-DVB-GMA-PEGMA (d). Full spectra (**left**); Detail of the FT-IR in the range 3350–3600 cm^−1^ (**right**).

**Figure 2 materials-16-02999-f002:**
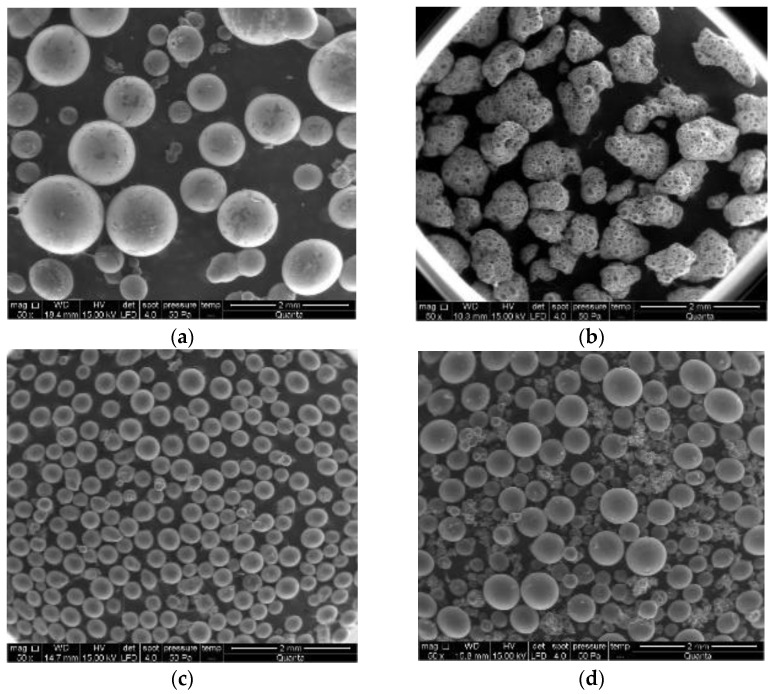
SEM micrographs of the different synthesized particles: St-DVB (**a**), St-DVB-PEGMA (**b**), St-DVB-GMA (**c**), and St-DVB-GMA-PEGMA (**d**).

**Figure 3 materials-16-02999-f003:**
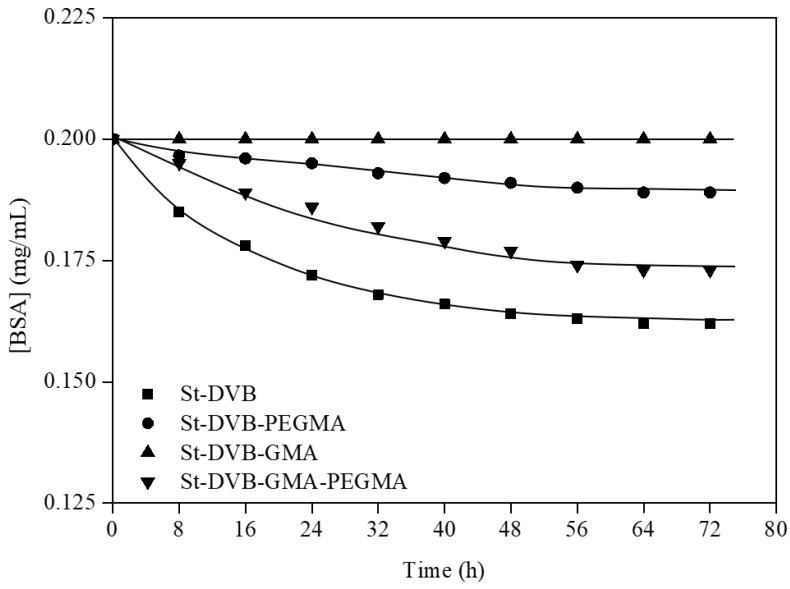
Kinetics of BSA immobilization in St-DVB non-pre-wetted particles. Wp = 2.5 g; T = 25 °C; t = 72 h and Fr = 8 mL/min.

**Figure 4 materials-16-02999-f004:**
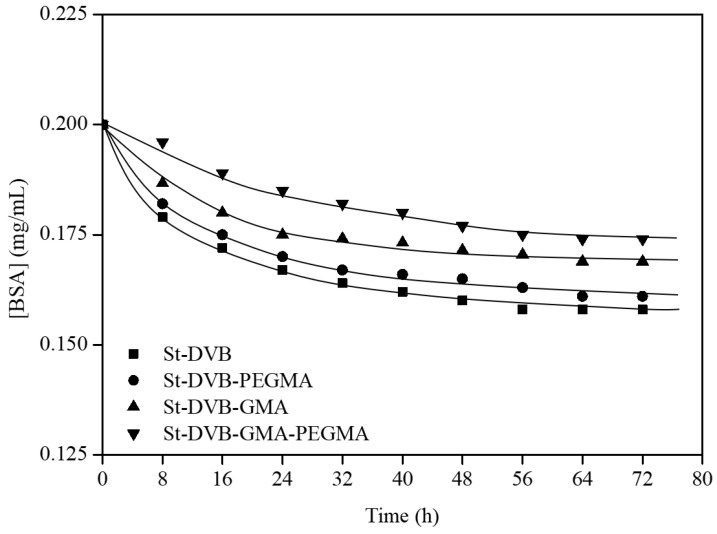
Kinetics of BSA immobilization in St-DVB pre-wetted particles. Wp = 2.5 g; T = 25 °C; t = 72 h and Fr = 8 mL/min.

**Figure 5 materials-16-02999-f005:**
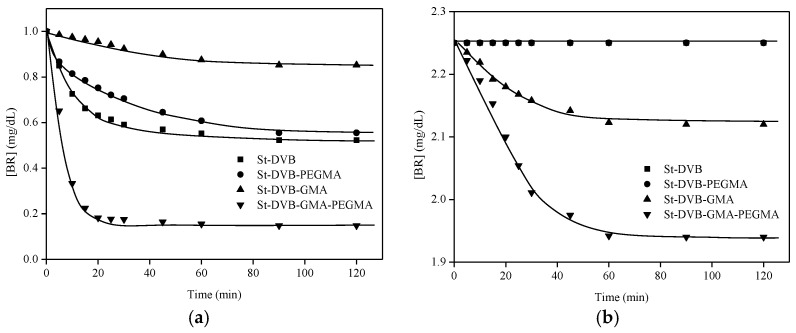
Kinetics of BR removal by non-pre-wetted particles without (**a**) and with BSA (**b**). Experiments without BSA were carried out with 7 g of particles and the experiments with BSA were carried out with 12 g of particles. T = 23 °C and Fr = 120 mL/min.

**Figure 6 materials-16-02999-f006:**
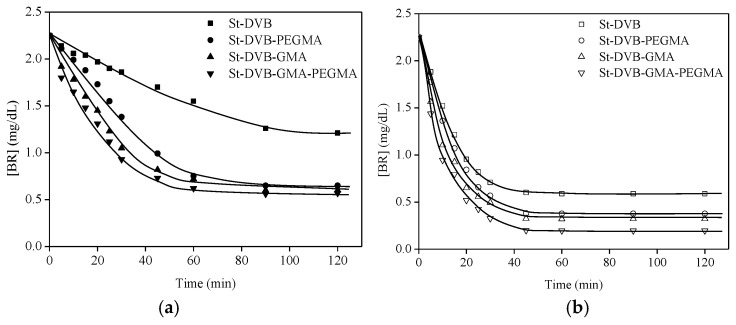
Kinetics of BR removal in St-DVB pre-wetted particles without (**a**) and with BSA (**b**). Experiments without BSA were carried out with 2 g of particles and the experiments with BSA were carried out with 1.5 g of particles. T = 25 °C and Fr = 120 mL/min.

**Figure 7 materials-16-02999-f007:**
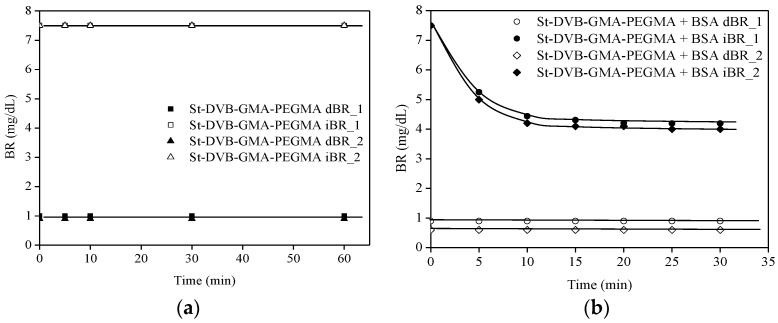
Kinetics of bilirubin removal from plasma by St-MMA-GMA-PEGMA particles without BSA (**a**) and with BSA (**b**) wetted with ethyl lactate. Wp = 1.2 g; T = 25 °C and Fr = 120 mL/min.

**Table 1 materials-16-02999-t001:** Reaction recipes for the synthesis of the St-DVB based particles.

Reagent	St-DVB	St-DVB-PEGMA	St-DVB-GMA	St-DVB-GMA-PEGMA
Milli-Q water	75.34	72.15	73.35	73.35
PVP	0.07	0.07	0.07	0.07
BPO	0.24	0.22	0.22	0.22
St	21.64	20.74	21.08	21.08
DVB	2.71	2.59	2.64	2.64
GMA	-	-	2.64	1.32
PEGMA	-	2.59	-	1.32

**Table 2 materials-16-02999-t002:** Median diameter in volume, Span, estimated pressure drops and yield.

Polymer	Dp_v 0.5_ (μm)	Span	ΔP (mmHg)	Yield (%)
St-DVB	868	1.1	2	96.5
St-DVB-PEGMA	769	0.9	4	60.8
St-DVB-GMA	460	1.0	9	87.3
St-DVB-GMA-PEGMA	505	1.2	7	87

**Table 3 materials-16-02999-t003:** Amount of removed bilirubin from PBS in particles without and with BSA wetted in ethyl lactate.

Particle	BR Uptake (mg/g)	
	Without BSA	With BSA
St-DVB	1.6	2.8
St-DVB-PEGMA	2.4	3.1
St-DVB-GMA	2.4	3.2
St-DVB-GMA-PEGMA	2.5	3.4

## Data Availability

All data generated or analyzed during this study are included in this article. Further enquiries can be directed to the corresponding author.

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
