# Peer review of "Crosslinked Bifunctional Particles for the Removal of Bilirubin in Hyperbilirubinemia Cases"

_materials, 2023, doi:10.3390/ma16082999_

Round 1

Reviewer 1 Report

Manuscript ID:   materials-2210358

Journal : Materials

Title: Crosslinked Bifunctional Particles for the Removal of Bilirubin 2 in Hyperbilirubinemia Cases

In this manuscript the authors have reported synthesis poly(styrene-co-divinylbenzene) as the base material and functionalized it with PEGMA and GMA to derive crosslinked material for the elimination of bilirubin. Although there are a quite few studies, however the proper explanation is absent. The writing part should be modified for better clarification. Therefore, major revision is needed prior approval for publication.

Decision:   Major revision

1. Abbreviations should be described properly prior using first throughout the manuscript. For instance, in the first line of the abstract and later on the authors have used several abbreviations without proper explanation. Prior using, they should be clarified properly.

2. The introduction section has majorly focused on previous literatures. The author should explain a bit more about their work so that readers can understand the summary of the work before reading the full manuscript.

3. From the experimental data, it is evident that the presence of albumin actually triggers the removal of bilirubin. However, the authors did not provide any concrete explanation about the cause of interaction. They should perform a control assay to assess the interaction between bilirubin and albumin that might provide insight.

Author Response

First of all, thanks for the time and effort in the manuscript revision and for your comments.

In this manuscript the authors have reported synthesis poly(styrene-co-divinylbenzene) as the base material and functionalized it with PEGMA and GMA to derive crosslinked material for the elimination of bilirubin. Although there are a quite few studies, however the proper explanation is absent. The writing part should be modified for better clarification. Therefore, major revision is needed prior approval for publication.

All the new descriptions and explanations are highlighted in yellow color.

  1. Abbreviations should be described properly prior using first throughout the manuscript. For instance, in the first line of the abstract and later on the authors have used several abbreviations without proper explanation. Prior using, they should be clarified properly.

The proper explanation of all the abbreviations used in the abstract have been included  prior the use in the abstract and the rest of the manuscript (Lines 15, 16, 17, 20, 83, 84 and 113).

  1. The introduction section has majorly focused on previous literatures. The author should explain a bit more about their work so that readers can understand the summary of the work before reading the full manuscript.

The novelty of this research is the synthesis of St-DVB particles functionalized with GMA and/or PEGMA brushes by suspension polymerization and the application of this particles to immobilized protein and then, remove bilirubin. The use of this brushes improves the capacity of immobilize albumin, while the use of DVB as crosslinker reduce the swelling effect described for St-MMA functionalized particles and avoiding by this way the coagulation problems caused by these particles when blood flows through the particles bed.

A brief description of the present work, highlighting the novelty or interest of this research has been included in Lines 78-86.

  1. From the experimental data, it is evident that the presence of albumin actually triggers the removal of bilirubin. However, the authors did not provide any concrete explanation about the cause of interaction. They should perform a control assay to assess the interaction between bilirubin and albumin that might provide insight.

The interaction of bilirubin and albumin in human body has been widely studied before. Due to its insolubility, bilirubin is needed to be linked to albumin to be transported to the liver by means of primary specific sites which have high selectivity and a linking constant of 1.4·107 L/mol at 37 ºC, and other secondary sites in albumin with lower selectivity.

This explanation has been included in Lines 41 and 43 as well as two references (3 and 4) where this fact is explained.

Reviewer 2 Report

María del Prado Garrido et al., have reported an interesting work on the development of St-DVB particles with PEGMA and/or GMA brushes for 15 the removal of bilirubin from blood in haemodialyzed patients. This is highly important and very crucial for the development of new materials that can remove bilirubin quickly and selectively. This technology is very helpful especially for the dialysis patients. The test of removal of bilirubin is really well planned and conducted results were noteworthy.

Figure 5. Experiments without BSA were carried out with 7 g of particles and the experiments with BSA were carried out with 12 g of particles. T = 23 °C and Fr = 120 mL/min. I would recommend authors to conduct the same experiment at human body temperature which is 37 °C  at a Flow rate of 3.0-26 ml/min as this is in arteries and and 1.2-4.8 ml/min in veins. It would more interesting to compare the results.

Author Response

María del Prado Garrido et al., have reported an interesting work on the development of St-DVB particles with PEGMA and/or GMA brushes for 15 the removal of bilirubin from blood in haemodialyzed patients. This is highly important and very crucial for the development of new materials that can remove bilirubin quickly and selectively. This technology is very helpful especially for the dialysis patients. The test of removal of bilirubin is really well planned and conducted results were noteworthy.

Figure 5. Experiments without BSA were carried out with 7 g of particles and the experiments with BSA were carried out with 12 g of particles. T = 23 °C and Fr = 120 mL/min. I would recommend authors to conduct the same experiment at human body temperature which is 37 °C at a Flow rate of 3.0-26 ml/min as this is in arteries and 1.2-4.8 ml/min in veins. It would be more interesting to compare the results.

Authors’ answers

First of all, thanks for the time and effort in the manuscript revision and for your positive comments.

We agree in the interest of performing the experiments in conditions close to human body, but we prefer doing it in conditions of the dialyses treatments, since the aim is to place this material in line with the conventional cartridges. These devices work at flow rates between 100 and 150 mL/min for children above 20kg and even higher for adults. On the other hand, the temperature in the line and cartridge is not 37 ºC, in fact, and after passing by the cartridge, the blood can be passed throughout a device for temperature increase in order to reach the corporeal temperature before entering in the body avoiding the hypothermia of the patient. To clarify these issues, the protocol for extracorporeal treatments has been included in the references of the manuscript in Line 128 (reference: M. José Santiago Lozano Jesús López-Herce Cid Lorena Bermúdez Barrezueta Sylvia Belda Hofheinz CARGO, “PROTOCOLO DE TÉCNICAS DE DEPURACIÓN EXTRARRENAL" Junio 2020).  

Reviewer 3 Report

1.      This work done in this manuscript describes the development of St-DVB particles with PEGMA and/or GMA brushes for the removal of bilirubin from blood in haemodialyzed patients, what could be the possible route of administration of such significantly larger particles?

2.      Are the sizes of the developed particles as mentioned in Table 2, suitable for their intended use?

3.      As mentioned in section 3.1: “These findings suggest that functionalization with GMA has a significant effect on particle size reduction”, what were actual values of reduced size of the developed nanocarriers? Their actual values should be mentioned here.

4.      Are the values of the distribution width (Span) as mentioned in Table are acceptable?

5.      For the intended application of the developed particles, the surface charges on the particle might play a major role in their actual functioning, so the zeta potential of the developed particles should be measured. Also the effect of surface charges/ zeta potential should be discussed in relation to their bilirubin removal activities.

6.      The actual pH of the used Phosphate Buffered Saline (PBS) should be mentioned in the texts of the manuscript.

7.      As per the title “Crosslinked Bifunctional Particles for the Removal of Bilirubin 2 in Hyperbilirubinemia Cases” of this article, there must be some experimental works to be done in vivo to justify this.

8.      The kinetics of BSA immobilization in St-DVB pre-wetted particles as mentioned in Figure 3 and Figure 4, was these conducted up to 72 h? This should be properly described in the methodology section also.

Author Response

First of all, thanks for the time and effort in the manuscript revision and for your valuable comments.

  1. This work done in this manuscript describes the development of St-DVB particles with PEGMA and/or GMA brushes for the removal of bilirubin from blood in hemodialyzed patients, what could be the possible route of administration of such significantly larger particles?

As mentioned in the manuscript (Lines 51-59), there are some available hemodialysis treatments based on membranes, but they are not effective enough. To improve these alternatives, particulate materials has been synthesized in order to improve existing treatments. These particles would not be administered as a typical drug, they will be introduced in a traditional hemodialysis cartridge for the use as hyperbilirubinemia treatment through an extracorporeal treatment in which the blood is pumped to get in contact with the particles.

In order to clarify this issue, a short explanation about the use of this material has been included in Lines 80-87.

  1. Are the sizes of the developed particles as mentioned in Table 2, suitable for their intended use?

The particle size is a key factor and it is important to reach a compromise between the improvement of albumin immobilization, that usually increases with the particle decrease, and the increase in the pressure drop when small particle size is used.

Thus, since the pressure drop is a key factor of this application because high pressure drops can coagulate the blood during the flow of blood by the particles bed, the pressure drop caused by a particles bed with similar dimensions of hemodialysis cartridges (that means a length of 25 cm and a diameter of 3 cm) has been theoretically determined by Ergun equation and these values are summarized in Table 2. These values are lower than the maximum pressure drop admissible for extracorporeal systems (200 mmHg). This information has been now included in line 177. Thus, it can be stated that the particle size is suitable for the application of bilirubin removal.

  1. As mentioned in section 3.1: “These findings suggest that functionalization with GMA has a significant effect on particle size reduction”, what were actual values of reduced size of the developed nanocarriers? Their actual values should be mentioned here.

The use of GMA functionalization and PEGMA-GMA functionalization resulted in a reduction in the median particle size of 53% and 42% respect to St-DVB non functionalized particles. The values of the particle size as well as the percentage reduction of the particle size are included in Table 2 and Lines 169-171 respectively.

  1. Are the values of the distribution width (Span) as mentioned in Table 2 are acceptable?

      It is important to consider that suspension polymerization process implies the obtention of a not very wide particle size distribution, a narrower PSD is not achievable with this method. The Span values close to one means an acceptable particle size distribution in which number and volume median particle size are similar, demonstrating the uniformity of the synthesized particles. In this case, all the values are close to one, so can be considered acceptable.

  1. For the intended application of the developed particles, the surface charges on the particle might play a major role in their actual functioning, so the zeta potential of the developed particles should be measured. Also, the effect of surface charges/ zeta potential should be discussed in relation to their bilirubin removal activities.

Good advice the indication to check the zeta potential of the synthesized particles. Nevertheless, it is important to remind that the objective of this research is the irreversible attachment of albumin to the particles by means of covalent link between amino acid groups present in the albumin and epoxide groups present in glycidyl methacrylate brushes incorporated in the particles, so the covalent interaction finish being stronger than the electrostatic interaction as has been previously described in literature. On the other hand the bilirubin uptake by the albumin is due to the high affinity primary sites (with a constant of 1.4·107 L/mol as mentioned in Lines 41-43) present in albumin, and not only by charges interaction.

On the other hand, it is worthy to point out that while there is no theoretical limit to the upper particle size range of zeta potential measurements, in reality the analysis becomes more difficult in the presence of gravitational settling. Measuring over 100 µm particles is extremely difficult and required manual manipulation in order to generate fairly repeatable results. The diameter desired and mostly obtained in our experiments for a good flow of the blood through the cartridge was about 500 microns. The manufacturers of zeta potential measuring devices don’t recommend measure samples this large for zeta potential. In fact our group has a zeta potential apparatus in our lab but the provided did not recommend us to use for these particles.

  1. The actual pH of the used Phosphate Buffered Saline (PBS) should be mentioned in the texts of the manuscript.

      Phosphate Buffered Saline (PBS) with concentration 0.01M and a pH of 7.4 is used for the experiments. This information is provided in Section 2.1. Materials (Line 96).

  1. As per the title “Crosslinked Bifunctional Particles for the Removal of Bilirubin in Hyperbilirubinemia Cases” of this article, there must be some experimental works to be done in vivo to justify this.

Besides, we consider important to include in the title a reference to the final application since the development has been guided for the requirements for the hyperbilirubinemia treatment. The concentration of bilirubin in PBS or plasma, the particle size, the polymers compatibility with blood, the use of BSA, the used flux and pH, the final use of plasma…all obey to the final use in the treatment of hyperbilirubinemia cases. A reference to the considered protocol for the variables’ selection, which is the protocol for extracorporeal treatments, has been included in the references of the manuscript in Line 128 (reference: M. José Santiago Lozano Jesús López-Herce Cid Lorena Bermúdez Barrezueta Sylvia Belda Hofheinz CARGO, “PROTOCOLO DE TÉCNICAS DE DEPURACIÓN EXTRARRENAL" Junio 2020) to support this idea.   

Several batches of particles were prepared and placed in cartridges to make “in-vivo” experiments with mini-pigs, based in the conditions reported in this work. Firstly, was needed to develop a proper experimental model to induce a stable hyperbilirubinemia in the experimentation animal. Once gotten a robust and convinced experimental model for the induction of hyperbilirubinemia, several experiments with extracorporeal treatment of the pig blood were done with successful results. The experimental model for the induction of the hyperbilirubinemia before the extracorporeal treatment is about to be published including the results of the treatment with the cartridges and we have considered not appropriate to give these results in advance.

We can state at this moment that the use of plasma with a concentration of bilirubin in the range of those from patients with hyperbilirubinemia throws results quite similar to in vivo experiments. For instance, the use of PBS did not demonstrate the selectivity reached with the PBS incorporation, while those in plasma showed it clearly.

  1. The kinetics of BSA immobilization in St-DVB pre-wetted particles as mentioned in Figure 3 and Figure 4, was these conducted up to 72 h? This should be properly described in the methodology section also.

      The duration of the experiments for albumin immobilization is described in the section 2.2.2. Albumin immobilization (Lines 121 and 122). However, the time required for this experiment has been also included in the captions of Figures 3 and 4 (Lines 225 and 237).

Round 2

Reviewer 1 Report

I have checked the response file and the revised manuscript. Based on their revision, I am satisfied that they have answered the queries. In my opinion, the manuscript in its present form is acceptable for publication.

Reviewer 3 Report

The authors have revised the manuscript well.